# COVID-19: Factors Associated with the Psychological Distress, Fear and Resilient Coping Strategies among Community Members in Saudi Arabia

**DOI:** 10.3390/healthcare11081184

**Published:** 2023-04-20

**Authors:** Talal Ali F. Alharbi, Alaa Ashraf Bagader Alqurashi, Ilias Mahmud, Rayan Jafnan Alharbi, Sheikh Mohammed Shariful Islam, Sami Almustanyir, Ahmed Essam Maklad, Ahmad AlSarraj, Lujain Nedhal Mughaiss, Jaffar A. Al-Tawfiq, Ahmed Ali Ahmed, Mazin Barry, Sherief Ghozy, Lulwah Ibrahim Alabdan, Sheikh M. Alif, Farhana Sultana, Masudus Salehin, Biswajit Banik, Wendy Cross, Muhammad Aziz Rahman

**Affiliations:** 1Department of Community, Psychiatric and Mental Health Nursing, College of Nursing, Qassim University, Buraidah 51452, Saudi Arabia; 2Department of Epidemiology and Public Health, Riyadh Second Health Cluster, King Fahad Medical City, Riyadh 12271, Saudi Arabia; aaalqurashi@kfmc.med.sa; 3Department of Public Health, College of Public Health and Health Informatics, Qassim University, Al Bukairiyah 52741, Saudi Arabia; 4Department of Emergency Medical Service, Jazan University, Jazan 45142, Saudi Arabia; 5Central Clinical School, Faculty of Medicine Nursing and Health Sciences, Monash University, Melbourne, VIC 3004, Australia; 6Institute for Physical Activity and Nutrition, School of Exercise and Nutrition Sciences, Deakin University, Melbourne, VIC 3125, Australia; 7Department of Medicine, Ministry of Health, Riyadh 3125, Saudi Arabia; 8College of Medicine, Alfaisal University, Riyadh 50927, Saudi Arabia; 9College of Medicine, Dar al Uloom University, Riyadh 7222, Saudi Arabia; 10Specialty Internal Medicine and Quality Department, Johns Hopkins Aramco Healthcare, Dhahran 34465, Saudi Arabia; 11Infectious Disease Division, Department of Medicine, Indiana University School of Medicine, Indianapolis, IN 47405, USA; 12Infectious Disease Division, Department of Medicine, Johns Hopkins University School of Medicine, Baltimore, MD 21205, USA; 13Faculty of Pharmacy, Alexandria University, Alexandria 21523, Egypt; 14Division of Infectious Diseases, Department of Internal Medicine, College of Medicine, King Saud University, Riyadh 11421, Saudi Arabia; 15Radiology Department, Mayo Clinic, Rochester, MN 55902, USA; 16Infectious Diseases Department, King Saud Medical City, Riyadh 11421, Saudi Arabia; 17School of Public Health and Preventive Medicine, Monash University, Melbourne, VIC 3004, Australia; 18Telstra Health, Melbourne, VIC 3002, Australia; 19Institute of Health and Wellbeing, Federation University Australia, Berwick, VIC 3806, Australia; 20Manna Institute, Mental Health Research and Training for Regional Australia, Regional University Network (RUN), The University of New England, Armidale, NSW 2351, Australia; 21Australian Institute for Primary Care & Ageing (AIPCA), La Trobe University, Melbourne, VIC 3086, Australia; 22Department of Noncommunicable Diseases, Bangladesh University of Health Sciences (BUHS), Dhaka 1704, Bangladesh; 23Faculty of Public Health, Universitas Airlangga, Surabaya 60115, Indonesia

**Keywords:** COVID-19, community, mental health, psychological distress, coping, resilience

## Abstract

(1) Background: COVID-19 caused the worst international public health crisis, accompanied by major global economic downturns and mass-scale job losses, which impacted the psychosocial wellbeing of the worldwide population, including Saudi Arabia. Evidence of the high-risk groups impacted by the pandemic has been non-existent in Saudi Arabia. Therefore, this study examined factors associated with psychosocial distress, fear of COVID-19 and coping strategies among the general population in Saudi Arabia. (2) Methods: A cross-sectional study was conducted in healthcare and community settings in the Saudi Arabia using an anonymous online questionnaire. The Kessler Psychological Distress Scale (K-10), Fear of COVID-19 Scale (FCV-19S) and Brief Resilient Coping Scale (BRCS) were used to assess psychological distress, fear and coping strategies, respectively. Multivariate logistic regressions were used, and an Adjusted Odds Ratio (AOR) with 95% Confidence Intervals (CIs) was reported. (3) Results: Among 803 participants, 70% (*n* = 556) were females, and the median age was 27 years; 35% (*n* = 278) were frontline or essential service workers; and 24% (*n* = 195) reported comorbid conditions including mental health illness. Of the respondents, 175 (21.8%) and 207 (25.8%) reported high and very high psychological distress, respectively. Factors associated with moderate to high levels of psychological distress were: youth, females, non-Saudi nationals, those experiencing a change in employment or a negative financial impact, having comorbidities, and current smoking. A high level of fear was reported by 89 participants (11.1%), and this was associated with being ex-smokers (3.72, 1.14–12.14, 0.029) and changes in employment (3.42, 1.91–6.11, 0.000). A high resilience was reported by 115 participants (14.3%), and 333 participants (41.5%) had medium resilience. Financial impact and contact with known/suspected cases (1.63, 1.12–2.38, 0.011) were associated with low, medium, to high resilient coping. (4) Conclusions: People in Saudi Arabia were at a higher risk of psychosocial distress along with medium-high resilience during the COVID-19 pandemic, warranting urgent attention from healthcare providers and policymakers to provide specific mental health support strategies for their current wellbeing and to avoid a post-pandemic mental health crisis.

## 1. Introduction

The COVID-19 disease, since its outbreak in China, has spread widely, affecting more than 213 countries and territories around the world. As of Dec-2021, globally, more than 276 million people have tested positive for COVID-19 infection, with more than five million fatalities [1]. COVID-19 caused the worst international health crisis of recent times, accompanied by a major global economic downturn and mass-scale job losses, with all of these impacts leading to psychosocial issues among people. Countries were forced to adopt extreme measures such as quarantine or self-isolation of the infected and their close contacts, preventing public gatherings, closing schools and universities, banning travel, closing territorial and international borders and in some cases, forcing complete lockdown of cities [2].

In Saudi Arabia, the first COVID-19-positive case was identified in the first week of March 2020 [3]. As of April 2022, Saudi Arabia has recorded 751,404 cases, with 9053 fatalities [1]. The lower fatality rate of 1.6% in Saudi Arabia compared with the international rate of 1.9% indicates that Saudi Arabia has handled the crisis relatively well compared to other countries [4]. The authorities in Saudi Arabia responded to the pandemic rapidly and imposed several measures to reduce the spread of the infection, closed all borders and suspended international flights and internal transports, including pilgrimages (Pilgrimages: in Islamic terminology, Hajj is a pilgrimage made to the Kaaba, the “House of God”, in the sacred city of Mecca in Saudi Arabia) to the Prophet’s Mosque in Madinah, an unprecedented decision since 1858 [5]. Curfew was imposed for several hours a day in many of the cities, together with the closure of schools and workplaces and the cancellation of larger social and religious events and services [5]. Furthermore, a national campaign of mass screening was initiated, where people with COVID-19-like symptoms were screened, along with their close contacts [6]. Those who returned a positive Polymerase Chain Reaction (PCR) test had to undergo mandatory 14-day quarantine [7]. In addition, arriving travelers were initially required to undergo institutional quarantine.

Saudi citizens and visitor residents were being regularly updated with the latest news and preventative measures by text messages [7,8]. The supply of essential goods such as food and medicines was ensured by home delivery.

Reports from several countries have indicated that the drastic but unavoidable measures that were taken to prevent the spread of COVID-19 have deeply impacted people’s lifestyles, with negative physical and mental health consequences [8,9,10]. There has been widespread anxiety and distress in all affected countries arising from prolonged isolation and quarantine, infection fears, frustration, boredom, shortage of essential supplies, inadequate information, and financial losses [11]. Even in countries such as Australia, where the infection and case fatality rates were very low compared to other developed countries such as USA or UK, people were distressed because of the potential of becoming infected with the virus, even without close contact with an infected person and the rapid spread within the communities [12,13]. Several research studies from various countries, including Saudi Arabia, found that the pandemic caused increased psychological distress, fear, and depression in a large proportion of community members [14].

Depoux et al. [15] warned of the possibility of adverse psychosomatic outcomes among people due to the pandemic, which is likely to be compounded by the constant flow of information (sometimes misinformation) via online and various forms of social media. It is feared that the rapidly expanding mass hysteria and panic regarding COVID-19 may lead to long-term psychological problems among people, regardless of their socioeconomic status [15]. The limited studies on the impact of pandemics on society, in relation to previous experiences such as SARS, have pointed to many stressors linked to disease outbreaks and pandemics [16,17]. One study from Bahrain examined the impact of COVID-19 on a fraction of individuals who were isolated or quarantined [18]. A study from UAE examined such impacts among university students only, and an additional study from Saudi Arabia examined the effect of lockdown on the psychology of the studied population [19]. Few studies have examined factors associated with mental wellbeing within Saudi Arabia, and no study focused on the psychological impact of COVID-19 [20]. A Saudi survey reported that of 3017 respondents, 19.6% had moderate to severe levels of anxiety during the pandemic [21]. Another study reported the occurrence of moderate or severe psychological impact among 23.6% of respondents [22]. Regarding residents in Saudi Arabia, the majority of them are temporary workers, particularly from low-income and middle-income countries. Previous research from Saudi Arabia examined the psychological impact on healthcare workers [22,23]. However, very limited evidence was generated amongst the general population in Saudi Arabia, and identification of high-risk groups during the pandemic was almost non-existent. Variations in COVID-19 restrictions, available resources, and compliance with public health messages made it difficult to apply the study findings from the neighboring countries to Saudi settings. Findings from this study would inform potential interventions and policies to address psychological wellbeing amongst the identified high-risk group of people in Saudi Arabia, specifically during similar crisis moments. This study examined factors associated with psychosocial distress, fear of COVID-19 and coping strategies amongst the general population in Saudi Arabia and defined high-risk groups impacted by the pandemic.

## 2. Materials and Methods

### 2.1. Study Design and Settings

This cross-sectional study was conducted among the general population in Saudi Arabia. The study was conducted over 30 days, from 15 December 2020 to 15 January 2021. The participants were aged between 18 and 65 years old. The questionnaire was designed in accordance with the previously published literature, and the survey was pre-tested for validation amongst migrants and non-migrants [13,24]. The survey was conducted in Arabic and English and took about 15 min to be completed.

### 2.2. Study Population and Sample Size

The study population included people residing in Saudi Arabia (irrespective of nationality), ≥18 years old, who could respond to either Arabic or English questionnaires on an online platform. This included patients, frontline health and other essential service workers, and general community members. Any participant who took <1 min to respond to the questionnaire was excluded from analyses due to the unreliability of responses. Snowball sampling was used to select study participants so that the respondents who were directly contacted by the study investigators could forward the survey link to their personal and professional networks. OpenEpi software v.3.01 was used to calculate the sample size. Considering Saudi Arabia’s population of 35.3 million (Worldometer, 2021), assuming a 23.6% prevalence of stress among the people at 95% confidence intervals and 80% power [25], the estimated minimum sample size was 278.

### 2.3. Data Collection

An online link in Arabic and English was created with a structured survey questionnaire using Google Forms. The plain language information statement and the consent form appeared on the first screen. Only the participants, who provided consent and agreed to participate in the study, could move to the next screen containing the single eligibility criteria of being an adult. The subsequent screens had the complete study questionnaire. The anonymous questionnaire was introduced, and the invitation, which included an internet link and a QR code, was distributed via social media platforms, online community networks, and staff and student email databases of participating universities/hospitals. Participants had the freedom to complete the questionnaire in their free time at home or while waiting to see a doctor. The online survey did not capture any personally identifiable information from them.

### 2.4. Study Tool

The structured survey questionnaire was adapted from the previous Australian study that was conducted with the same objective as this study [13]. Psychological distress was measured using the Kessler Psychological Distress Scale (K-10) [26]. The K10 scale is a self-reporting questionnaire with ten items that assess distress based on depressive and anxiety symptoms. Each item has five possible responses (none of the time = 1, some of the time = 3, most of the time = 4, all of the time = 5). The fear was measured using the Fear of COVID-19 Scale (FCV-19S) [27]. The FCV-19S is a seven-item scale that measures general community fear of COVID-19. There are five alternative responses for each item strongly disagree = 1, disagree = 2, neutral = 3, agree = 4, strongly agree = 5, etc.) There are five alternative responses for each issue, and coping was measured using Brief Resilient Coping Scale (BRCS) [28]. The BRCS is a four-item scale that assesses resilience, a component of psychological wellness. Each question has a five-point scale, with “does not describe me at all = 1”, “does not describe me = 2”, “neutral = 3”, “describes me = 4”, “describes me very well = 5”. The entire questionnaire, including the study tools, was translated and back-translated from English to Arabic by two independent translators, which were then verified by the study investigators. While the Arabic version of validated K-10 and FCV-19S were readily available to use, the Arabic version of the BRCS was not available, which was translated following the process mentioned above. The tools have been recently examined for reliability and validity, and it was found that these tools are valid and reliable amongst both migrant and non-migrant populations in Australia [24]. The reliability of those tools, measured as Cronbach’s alpha, showed satisfactory performance in this study (K-10: 0.94, FCV-19S: 0.89, BRCS: 0.74).

### 2.5. Ethical Considerations

All participants were requested to sign a consent form before filling out the questionnaire to register their willingness to participate. All methods were performed in accordance with the relevant guidelines and regulations of the Declaration of Helsinki. Ethical approval was obtained from the Human Research Ethics Committee in King Fahad Medical City, Saudi Arabia (H-01-R-012).

### 2.6. Data Analyses

The database was downloaded from Google Forms and analyzed using SPSS v.25. Descriptive analyses were undertaken to describe the study variables. Study outcomes were psychological distress, fear of COVID-19, and coping. High-risk groups were identified by analyzing all the exposure variables data collected in this study. Median values were computed for the continuous variable (age) and each scale (K10, FCV-19S and BRCS) as none of those variables were normally distributed, evidenced by the Kolmogorov–Smirnov test (*p* > 0.05). Proportions were reported for categorical variables. To conduct inferential analysis, K-10 was defined into low (score 10–15) and moderate to very high (score 16–50), FCV-19S scale was categorized into low (score 7–21) or high (score 22–35) and BRCS into low (score 4–13) and medium to high (score 14–20) resilient coping (14). Cross-tabulation of the factors associated with psychological distress was done by comparing low and moderate to very high distress on the K-10 scale. Factors associated with fear of COVID-19 were identified by comparing low and high fear on the FCV-19S scale, and factors associated with coping were identified by comparing low and medium to high resilient coping on the BRCS scale. Multivariable logistic regression analyses (fulfilling the assumptions of independence of errors, linearity for the continuous variables, absence of multicollinearity and lack of strongly influential outliers) were performed to investigate the factors of moderate to very high distress on the K10 scale, the high level of fear of COVID-19 on the FCV-19S scale, and medium to high resilient coping on the BRCS scale. Statistical significance was determined by *p* < 0.05. Odds Ratios (ORs), with a 95% confidence interval (CI) used to assess the strength of the association. Adjusted ORs (AORs) indicated adjustment of potential confounding variables.

## 3. Results

A total of 803 individuals aged ≥18 years living in Saudi Arabia participated in this study. Table 1 presents the characteristics of the participants. More than half of the participants (57.1%) were 18–29 years, and the majority (69.5%) were females, had a bachelor’s degree or above (64.5%), and were living with family (85.8%). More than a third (34.6%) of the participants worked as frontline or essential service workers during the pandemic. Just over a third (33.7%) reported negative financial impact due to COVID-19. Only 16.3% of the participants were current smokers, and more than half (56.5%) of them increased smoking during the pandemic. About a quarter (24.3%) reported pre-existing comorbid conditions, including mental health issues (5.6%). About a tenth (8.2%) of the participants had tested positive for COVID-19, while over a tenth (12%) tested negative. More than a third (36.2%) had close contact with confirmed or suspected COVID-19 cases.

### 3.1. Psychological Distress

Among the study participants, 72% experienced moderate to very high levels of psychological distress (Table 2). After adjusting for the effects of potential confounders, evidence of significant association for moderate to very high psychological distress was observed with age, sex, nationality, perceived distress due to change of employment, the financial impact of COVID-19, having co-morbidities and current smoking (Table 3).

### 3.2. Fear of COVID-19

Among the participants, one in 10 (11.1%) demonstrated high levels of fear of COVID-19 (Table 4). In the multivariate analyses, it was found that a high level of fear was significantly associated with perceived distress due to changes in employment situations and smoking status. Individuals with moderate to a great deal of distress due to change in employment were more likely to experience high levels of fear of COVID-19 of 3.42 (95% CI: 1.91–6.11, *p* < 0.001) compared to individuals who perceived little or no distress. Being an ex-smoker was associated with higher levels of anxiety about COVID-19 compared to those who never smoked (AOR 0.72, 95% CI: 1.14–12.14, *p* = 0.029) (Table 5).

### 3.3. Coping Strategies

Just more than half (55.8%) of the participants had medium to high resilient coping (Table 6). Significant association for high resilience coping was observed with perceived distress due to changes in employment conditions, the economic impact of the COVID-19 pandemic, and contact with known or suspected cases. Individuals who reported a positive (AOR: 1.68, 95% CI: 1.03–2.75, *p* = 0.038) or negative economic impact (AOR: 1.82, 95% CI: 1.22–2.71, *p* = 0.003) of COVID-19 were more likely to have medium to high resilient coping. In addition, individuals who had contact with confirmed or suspected COVID-19 cases were more likely to have medium to high resilient coping (AOR: 1.63, 95% CI: 1.12–2.38, *p* = 0.011). On the other hand, those who perceived distress due to employment changes had low resilient coping (AOR 0.63, 95% CI 0.43–0.92, *p* = 0.017) (Table 7).

## 4. Discussion

This cross-sectional survey was conducted among people in Saudi Arabia. Aspects of psychological distress, fear of COVID-19, and coping strategies were assessed using K10, FCV-19S, and BRCS scales, respectively.

The current study indicated a high percentage (70%) of people who suffered from distress during the pandemic, the prevalence of which was more than double compared to other local research in Saudi Arabia. Alkhamees et al. [29] assessed psychological impact during an early stage of the pandemic and showed that a quarter of the participants suffered from moderate to severe psychological impact. Another study [21] conducted in May 2020 showed 40% of the general public in Saudi Arabia suffered from psychological distress caused by COVID-19. Thus, as COVID-19 lasted for a prolonged period, more people are expected to have a psychological impact, and more efforts are needed for psychological support. The same observation was noted in a Canadian study that showed a significant increase in stress during the COVID-19 outbreak [30]. A systematic review showed high levels of psychological distress during the COVID-19 period, with variable rates of anxiety, depression, and stress [31]. In addition, other factors may play a role in contributing to the increased level of psychological distress in this study, as the previous studies were conducted during the initial months of the pandemic. ‘Infodemic’ could potentially contribute to the heightened distress in this study, which requires further investigations.

In terms of associated factors, age, gender, nationality, perceived distress due to change of employment conditions, the financial impact of COVID-19, and smoking were significantly associated with higher levels of psychological distress. Similar to this study, research conducted in the US during the pandemic showed that women, Hispanics, Asians, families with children under 18, and foreign-born respondents had higher subjective fear and worry levels than their counterparts [30].

Individuals aged 18–29 years had higher psychological distress. This result coincides with a report from over 60 countries that found that younger age groups were more vulnerable to the mental health impact of the pandemic [32]. One explanation of the result could be dependence on inauthentic information received from social media platforms. Marar et al. [33] reported that most of the Saudi population used social media platforms when they needed health information. Another study showed that social media had a positive impact on the knowledge of the Saudi population about COVID-19 [34]. However, it was found that younger individuals were less likely to practice coping methods such as spirituality and mindfulness, which has proved to be a handy tool to control stress and depression [35].

In this study, smoking was associated with an AOR of 2.87 of psychological distress during the COVID-19 pandemic. According to evidence, smoking could cause symptoms like depression and anxiety [36]. In a study from England, there was a significant association between psychological distress and past smoking [37]. The study showed further deterioration in mental health among smokers during the COVID-19 pandemic. A systematic review showed a bi-directional effect between psychological distress and smoking [36]. Significantly, research finding indicates that 25% reported increasing smoking more than usual, and 51% smoked the same amount during the COVID-19 pandemic [38]. It is important to note that a recent study from Saudi Arabia showed a prevalence of cigarette smoking of 21.4% of the population [39]. Thus, it is essential to have further studies to ameliorate the risk of smoking and mental health in the population, especially during the pandemic.

Additionally, changes in employment conditions and financial challenges were related to a high level of fear from COVID-19 and psychological distress. The economic effect of COVID-19 was well-described worldwide [40]. In particular, research undertaken in Italy, India, South Africa, the UK and the USA identified that cigarette smokers bought more cigarettes than usual triggered by the fear that stores might run out of stock or be closed because of lockdowns during the pandemic [41].

In terms of coping strategies, more than half of the study participants were medium to high resilient copers. According to previous literature, the Saudis have been found to be quite resilient to COVID-19 stress in comparison to other countries experiencing this pandemic with high quality of life scores [42]. Unlike other places around the world suffering from lack of food and free treatment, unavailable beds in intensive care units, and an insufficient number of doctors, the Saudi government made an extraordinary effort in several areas economics, health, religious, social support, food, and quality of life [3].

Individuals in this research who came in contact with confirmed or suspected COVID-19 cases were more likely to have medium to high resilient coping, which could be due to accessing free treatment and an advanced healthcare system in Saudi. In addition, there is a clear relationship between coping strategy and stress outcome [43]. In one study, religion was one of the most frequent coping strategies among nursing students in Saudi Arabia [44]. An additional study among healthcare workers in Saudi Arabia showed increased stress levels with a considerable drop in resilient coping scores [45].

Saudi Arabia’s Ministry of Health has developed multiple methods to support the wellbeing and mental health of healthcare providers. Smartphone applications, hotlines and committees were available to tackle concerns and worries. It is important to note that different professions and age groups may use different coping strategies. In one study, nurses used avoiding coping styles and positive reappraisal more than doctors: those more than 40 years of age used social support, and those < 40 years of age relied on avoidance of stress management techniques [46]. Despite positive coping strategies in the health and medical fields, there were limited coping strategies for other essential service workers and general community members who suffered from COVID-19 and may have experienced psychological impacts. Future studies should focus on intervention measuring and programs among the general population in Saudi to identify coping strategies.

## 5. Study Limitations

This study had a large and representative sample from different categories of frontline workers and the general population and was conducted during the second wave of the pandemic. Findings will assist in having a clear vision for decision-makers to manage psychological distress and fear of COVID-19 with adaptable strategies for Saudi people. However, there are several limitations to this study. The use of an online self-administered questionnaire may have introduced response and recall bias. Additionally, the dissemination of questionnaires through social media platforms for recruitment resulted in having more participants from certain regions than others. The cross-sectional study design limited our ability to infer causations based on the identified statistical associations. In addition, snowball sampling limited the generalizability of study findings to the entire country. Therefore, findings need to be interpreted with caution. However, the study identified the high-risk groups of the population and provided earlier evidence amidst a critical pandemic period. A future large-scale study with a representative sample would validate our study findings. The presence of more females in the study population could be due to the online nature of the study, which does not truly reflect the population composition of Saudi Arabia. Further studies are also needed to address the evolution of the psychological impact of COVID-19 over time and to examine the post-COVID psychological impact in Saudi Arabia.

## 6. Conclusions

Identification of high-risk groups with increased psychological distress and fear during the current COVID-19 pandemic is critical. Factors identified in this study can strengthen illness prevention by guiding policymakers for such a vulnerable population. Healthcare authorities should monitor young people and smokers about their mental health, and considering a behavioral support program will be invaluable. Those affected by changes in employment and negative financial impacts should be prioritized within the current support services available in Saudi Arabia. Living in a COVID-safe environment and adopting a lifestyle supporting both physical and mental wellbeing during the pandemic era is warranted in Saudi Arabia.

## Figures and Tables

**Table 1 healthcare-11-01184-t001:** Characteristics of the study participants.

Characteristics	Total
	*n* (%)
Total study participants	803
Age in years	
Median	27
Range	18–61
Age groups	665
18–29 years	380 (57.1)
30–39 years	181 (27.2)
40–61 years	104 (15.6)
Gender	800
Male	244 (30.5)
Female	556 (69.5)
Educational attainment	800
Grade 1 to Grade 6	2 (0.3)
Grade 7 to Grade 12	203 (25.4)
Trade/Certificate/Diploma	79 (9.9)
Bachelor or above	516 (64.5)
Living status	800
Live without family members	114 (14.3)
Live with family members	686 (85.8)
Citizenship	800
Non-Saudi	122 (15.3)
Saudi	678 (84.8)
Current employment status	779
Unemployed	0
Jobs affected by COVID-19 (lost job/working hours reduced/	529 (67.9)
Jobs unaffected by COVID-19 (employed/Government benefits)	250 (32.1)
Perceived distress due to change of employment status	770
A little to none	497 (64.5)
Moderate to a great deal	273 (35.5)
Frontline or essential service worker (Self-identification)	803
No	525 (65.4)
Yes	278 (34.6)
COVID-19 impacted financial situation	803
No impact	399 (49.7)
Positive impact	133 (16.6)
Negative impact	271 (33.7)
Co-morbidities	803
No	608 (75.7)
Psychiatric/Mental health issues	45 (5.6)
Other co-morbidities *	150 (18.7)
Smoking	803
Never smoker	650 (80.9)
Ex-smoker (quit at least one month ago)	22 (2.7)
Current smoker (daily/non-daily/occasional)	131 (16.3)
Increased smoking over the last 6 months	131
No	57 (43.5)
Yes	74 (56.5)
Contact with known/suspected COVID-19 cases	784
No	500 (63.8)
Yes	284 (36.2)
COVID-19 related experiences	773
No known diagnosis of COVID-19	600 (77.6)
Tested positive for COVID-19	63 (8.2)
Tested negative for COVID-19 but self-isolating	93 (12.0)
Recent overseas travel history and was in quarantine	17 (2.2)
Healthcare service used to overcome COVID-19-related stress in the past 6 months	770
No	658 (85.5)
Yes	112 (14.5)

* Cardiac diseases/Stroke/Hypertension/Hyperlipidaemia/Diabetes/Cancer/Chronic respiratory illness.

**Table 2 healthcare-11-01184-t002:** Psychological distress among adults during the COVID-19 pandemic in Saudi Arabia.

Anxiety and Depression Checklist (K10)	Total
	*n* (%)
About how often did you feel tired for no good reason?	803
None	233 (29.0)
A little	193 (24.0)
Sometime	239 (29.8)
Most of the time	109 (13.6)
All the time	29 (3.6)
About how often did you feel nervous?	803
None	165 (20.5)
A little	218 (27.1)
Sometime	212 (26.4)
Most of the time	146 (18.2)
All the time	62 (7.7)
About how often did you feel so nervous that nothing could calm you down?	803
None	343 (42.7)
A little	178 (22.2)
Sometime	157 (19.6)
Most of the time	79 (9.8)
All the time	46 (5.7)
About how often did you feel hopeless?	803
None	329 (41.0)
A little	190 (23.7)
Sometime	139 (17.3)
Most of the time	95 (11.8)
All the time	50 (6.2)
About how often did you feel restless or fidgety?	803
None	185 (23.0)
A little	223 (27.8)
Sometime	225 (28)
Most of the time	136 (16.9)
All the time	34 (4.2)
About how often did you feel so restless you could not sit still?	803
None	333 (41.5)
A little	233 (29.0)
Sometime	140 (17.4)
Most of the time	77 (9.6)
All the time	20 (2.5)
About how often did you feel depressed?	803
None	310 (38.6)
A little	211 (26.3)
Sometime	153 (19.1)
Most of the time	90 (11.2)
All the time	39 (4.9)
About how often did you feel that everything was an effort?	803
None	227 (28.3)
A little	237 (29.5)
Sometime	168 (20.9)
Most of the time	89 (11.1)
All the time	82 (10.2)
About how often did you feel so sad that nothing could cheer you up?	803
None	286 (35.6)
A little	205 (25.5)
Sometime	159 (19.8)
Most of the time	92 (11.5)
All the time	61 (7.6)
About how often did you feel worthless?	803
None	417 (51.9)
A little	148 (18.4)
Sometime	105 (13.1)
Most of the time	71 (8.8)
All the time	62 (7.7)
K10 score (total)	803
Median	21
Minimum–Maximum	10 to 50
Level of psychological distress (K10 categories)	803
Low (score 10–15)	225 (28.0)
Moderate (score 16–21)	196 (24.4)
High (score 22–29)	175 (21.8)
Very high (score 30–50)	207 (25.8)

**Table 3 healthcare-11-01184-t003:** Factors associated with psychological distress among adults in Saudi Arabia.

Characteristics	*n* (Row %)			*p*	AOR	95% CI
	Low (Score 10–15)	Moderate to Very High (Score 16–50)	Total			
Age groups						
>29 years	118 (41.8)	164 (58.2)	282		1	
18–29 years	60 (15.8)	320 (84.2)	380	0.000	3.35	2.06–5.44
Sex						
Male	107 (43.9)	137 (56.1)	244		1	
Female	118 (21.2)	438 (78.8)	556	0.000	2.59	1.60–4.19
Educational attainment						
Grade 1–12	37 (18.0)	168 (82.0)	205		1	
Trade/Certificate/Diploma	36 (45.6)	43 (54.4)	79	0.138	0.51	0.21–1.24
Bachelor or above	150 (29.1)	366 (70.9)	516	0.875	0.95	0.53–1.71
Living status						
Live without family members	36 (31.6)	78 (68.4)	114			
Live with family members	187 (27.3)	499 (72.7)	686	0.119	1.7	0.87–3.32
Nationality						
Saudi	191 (28.2)	487 (71.8)	678		1	
Non-Saudi	33 (27.0)	89 (73.0)	122	0.024	2.17	1.11–4.26
Current employment condition						
Job unaffected by COVID-19 (employed/Government benefits)	56 (22.4)	194 (77.6)	250		1	
Job affected by COVID-19 (lost job/working hours reduced/	166 (31.4)	363 (68.6)	529	0.605	0.87	0.51–1.48
Perceived distress due to change of employment condition						
A little to none	181 (36.4)	316 (63.6)	497		1	
Moderate to a great deal	36 (13.2)	237 (86.8)	273	0.000	2.90	1.73–4.87
Frontline or essential service worker (self-identification)						
No	137 (26.1)	388 (73.9)	525		1	
Yes	88 (31.7)	190 (68.3)	278	0.778	0.93	0.57–1.52
COVID-19 impacted financial situation						
No impact	134 (33.6)	265 (66.4)	399		1	
Positive impact	37 (27.8)	96 (72.2)	133	0.133	1.55	0.87–2.76
Negative impact	54 (19.9)	217 (80.1)	271	0.003	2.14	1.29–3.56
Co-morbidities						
No	186 (30.6)	422 (69.4)	608		1	
Psychiatric/Mental health issues	4 (8.9)	41 (91.1)	45	0.091	2.72	0.85–8.66
Other co-morbidities	35 (23.3)	115 (76.7)	150	0.001	2.67	1.47–4.87
Smoking						
Never smoker	192 (29.5)	458 (70.5)	650		1	
Ex-smoker	8 (36.4)	14 (63.6)	22	0.782	1.18	0.37–3.79
Current smoker	25 (19.1)	106 (80.9)	131	0.001	2.87	1.55–5.33
Contact with known/suspected COVID-19 cases						
No	141 (28.2)	359 (71.8)	500		1	
Yes	77 (27.1)	207 (72.9)	284	0.184	0.73	0.46–1.16
COVID-19-related experiences						
No known diagnosis of COVID-19	165 (27.5)	435 (72.5)	600		1	
Tested positive for COVID-19	22 (34.9)	41 (65.1)	63	0.642	0.82	0.36–1.88
Tested negative for COVID-19 but self-isolating	26 (28.0)	67 (72.0)	93	0.612	0.84	0.43–1.64
Recent overseas travel history and was in quarantine	3 (17.6)	14 (82.4)	17	0.361	2.22	0.40–12.28

**Table 4 healthcare-11-01184-t004:** Fear of COVID-19 among adults in Saudi Arabia.

Fear of COVID-19 Scale (FCV-19S) Items	Total
	*n* (%)
I am most afraid of COVID-19	803
Strongly disagree	275 (34.2)
Somewhat disagree	175 (21.8)
Neither agree nor disagree	190 (23.7)
Somewhat agree	138 (17.2)
Strongly agree	25 (3.1)
It makes me uncomfortable to think about COVID-19	803
Strongly disagree	245 (30.5)
Somewhat disagree	162 (20.2)
Neither agree nor disagree	168 (20.9)
Somewhat agree	187 (23.3)
Strongly agree	41 (5.1)
My hands become clammy when I think about COVID-19	803
Strongly disagree	532 (66.3)
Somewhat disagree	160 (19.9)
Neither agree nor disagree	75 (9.3)
Somewhat agree	27 (3.4)
Strongly agree	9 (1.1)
I am afraid of losing my life because of COVID-19	803
Strongly disagree	388 (48.3)
Somewhat disagree	164 (20.4)
Neither agree nor disagree	127 (15.8)
Somewhat agree	84 (10.5)
Strongly agree	40 (5.0)
When watching news and stories about COVID-19 on social media, I become nervous or anxious	803
Strongly disagree	269 (33.5)
Somewhat disagree	166 (20.7)
Neither agree nor disagree	173 (21.5)
Somewhat agree	156 (19.4)
Strongly agree	39 (4.9)
I cannot sleep because I’m worrying about getting COVID-19	803
Strongly disagree	554 (69.0)
Somewhat disagree	130 (16.2)
Neither agree nor disagree	87 (10.8)
Somewhat agree	19 (2.4)
Strongly agree	13 (1.6)
My heart races or palpitates when I think about getting COVID-19	803
Strongly disagree	501 (62.4)
Somewhat disagree	145 (18.1)
Neither agree nor disagree	100 (12.5)
Somewhat agree	45 (5.6)
Strongly agree	12 (1.5)
FCV-19S score (total)	803
Median	13
Minimum–Maximum	7 to 35
Level of fear of COVID-19 (FCV-19S categories)	803
Low (score 7–21)	714 (88.9)
High (score 22–35)	89 (11.1)

**Table 5 healthcare-11-01184-t005:** Factors associated with the fear of COVID-19 among adults in Saudi Arabia.

Characteristics	*n* (Row %)			*p*	AOR	95% CI
	Low (Score 7–21)	High (Score 22–35)	Total			
Age groups						
18–29 years	339 (89.2)	41 (10.8)	380	0.989	1.00	0.53–1.88
>29 years	248 (87.0)	37 (13.0)	285	1.000		
Sex						
Male	220 (90.2)	24 (9.8)	244		1	
Female	492 (88.5)	64 (11.5)	556	0.602	1.20	0.61–2.38
Educational attainment						
Grade 1–12	189 (92.2)	16 (7.8)	205		1	
Trade/Certificate/Diploma	68 (86.1)	11 (13.9)	79	0.741	1.19	0.42–3.42
Bachelor or above	454 (88.0)	62 (12.0)	516	0.879	0.94	0.44–2.03
Living status						
Live without family members	90 (78.9)	24 (21.1)	114		1	
Live with family members	621 (90.5)	65 (9.5)	686	0.213	0.61	0.28–1.33
Nationality						
Saudi	608 (89.7)	70 (10.3)	678		1	
Non-Saudi	104 (85.2)	18 (14.8)	122	0.600	0.79	0.33–1.88
Current employment condition						
Job unaffected by COVID-19 (employed/Government benefits)	227 (90.8)	23 (9.2)	250		1	
Job affected by COVID-19 (lost job/working hours reduced/	464 (87.7)	65 (12.3)	529	0.188	1.58	0.80–3.12
Perceived distress due to change of employment condition						
A little to none	462 (93.0)	35 (7)	497			
Moderate to a great deal	219 (80.2)	54 (19.8)	273	0.000	3.42	1.91–6.11
Frontline or essential service worker (self-identification)						
No	489 (91.4)	45 (8.6)	525			
Yes	234 (84.2)	44 (15.8)	278	0.062	1.79	0.97–3.31
COVID-19 impacted financial situation						
No impact	366 (91.7)	33 (8.3)	399		1	
Positive impact	121 (91.0)	12 (9.0)	133	0.815	1.10	0.50–2.44
Negative impact	227 (83.8)	44 (16.2)	271	0.282	1.40	0.76–2.59
Co-morbidities						
No	544 (89.5)	64 (10.5)	608		1	
Psychiatric/Mental health issues	40 (88.9)	5 (11.1)	45	0.876	0.92	0.31–2.72
Other co-morbidities	130 (86.7)	20 (13.3)	150	0.207	1.55	0.78–3.07
Smoking						
Never smoker	587 (90.3)	63 (9.7)	650		1	
Ex-smoker	17 (77.3)	5 (22.7)	22	0.029	3.72	1.14–12.14
Current smoker	110 (84.0)	21 (16.0)	131	0.098	1.79	0.90–3.55
Contact with known/suspected COVID-19 cases						
No	454 (90.8)	46 (9.2)	500		1	
Yes	245 (86.3)	39 (13.7)	244	0.349	1.32	0.74–2.37
COVID-19 related experiences						
No known diagnosis of COVID-19	540 (90.0)	60 (10.0)	600			
Tested positive for COVID-19	55 (87.3)	8 (12.7)	63	0.263	0.54	0.19–1.58
Tested negative for COVID-19 but self-isolating	77 (82.8)	16 (17.2)	93	0.860	1.08	0.47–2.45
Recent overseas travel history and was in quarantine	12 (70.6)	5 (29.4)	17	0.564	0.52	0.06–4.84

**Table 6 healthcare-11-01184-t006:** Coping during the COVID-19 pandemic in Saudi Arabia.

Brief Resilient Coping Scale (BRCS)	Total
	*n* (%)
I look for creative ways to alter difficult situations	802
Does not describe me at all	87 (10.8)
Does not describe me	101 (12.6)
Neutral	278 (34.7)
Describes me	275 (34.3)
Describes me very well	61 (7.6)
Regardless of what happens to me, I believe I can control my reaction to it	802
Does not describe me at all	43 (5.4)
Does not describe me	82 (10.2)
Neutral	248 (30.9)
Describes me	332 (41.4)
Describes me very well	97 (12.1)
I believe I can grow in positive ways by dealing with difficult situations	802
Does not describe me at all	32 (4.0)
Does not describe me	42 (5.2)
Neutral	204 (25.4)
Describes me	395 (49.3)
Describes me very well	129 (16.1)
I actively look for ways to replace the losses I encounter in life	802
Does not describe me at all	63 (7.9)
Does not describe me	64 (8.0)
Neutral	252 (31.4)
Describes me	303 (37.8)
Describes me very well	120 (15.0)
BRCS score (total)	
Median	14
Minimum–Maximum	4 to 20
Levels of coping (BRCS categories)	
Low resilient copers (score 4–13)	354 (44.1)
Medium resilient copers (score 14–16)	333 (41.5)
High resilient copers (score 17–20)	115 (14.3)

**Table 7 healthcare-11-01184-t007:** Factors associated with coping with the COVID-19 pandemic in Saudi Arabia.

Characteristics	*n* (Row %)			*p*	AOR	95% CI
	Low Resilient (Score 4–13)	Medium to High Resilient (Score 14–20)	Total			
Age groups						
18–29 years	171 (45.1)	208 (54.9)	379	0.781	1.06	0.71–1.58
>29 years	122 (42.8)	163 (57.2)	285		1	
Sex						
Male	104 (42.8)	139 (57.2)	243		1	
Female	248 (44.6)	308 (55.4)	556	0.641	1.10	0.73–1.66
Educational attainment						
Grade 1–12	95 (46.6)	109 (53.4)	204		1	
Trade/Certificate/Diploma	39 (49.4)	40 (50.6)	79	0.643	1.19	0.58–2.44
Bachelor or above	217 (42.1)	299 (57.9)	516	0.201	1.33	0.86–2.07
Living status						
Live without family members	50 (43.9)	64 (56.1)	114		1	
Live with family members	301 (43.9)	384 (56.1)	685	0.510	0.83	0.47–1.46
Nationality						
Non-Saudi	54 (44.3)	68 (55.7)	122	0.271	0.73	0.42–1.27
Saudi	353 (44.2)	446 (55.8)	677		1	
Current employment condition						
Job unaffected by COVID-19 (employed/Government benefits)	109 (43.8)	140 (56.2)	249		1	
Job affected by COVID-19 (lost job/working hours reduced/	230 (43.5)	299 (56.5)	529	0.790	1.07	0.71–1.58
Perceived distress due to change of employment condition						
A little to none	217 (43.8)	279 (56.2)	496		1	
Moderate to a great deal	122 (44.7)	151 (55.3)	273	0.017	0.63	0.43–0.92
Frontline or essential service worker (self-identification)						
No	236 (45.0)	288 (55.0)	524		1	
Yes	118 (42.4)	160 (57.6)	278	0.519	1.14	0.76–1.71
COVID-19 impacted financial situation						
No impact	190 (47.6)	209 (52.4)	399		1	
Positive impact	51 (38.6)	81 (61.4)	132	0.038	1.68	1.03–2.75
Negative impact	113 (41.7)	158 (58.3)	271	0.003	1.82	1.22–2.71
Co-morbidities						
No	266 (43.8)	341 (56.2)	607		1	
Psychiatric/Mental health issues	22 (48.9)	23 (51.1)	45	0.363	0.72	0.37–1.46
Other co-morbidities	66 (44.0)	84 (56.0)	150	0.571	1.14	0.72–1.81
Smoking						
Never smoker	293 (45.1)	357 (54.9)	650		1	
Ex-smoker	7 (31.8)	15 (68.2)	22	0.353	1.64	0.58–4.66
Current smoker	54 (41.5)	76 (58.5)	130	0.610	1.13	0.71–1.79
Contact with known/suspected COVID-19 cases						
No	236 (47.3)	263 (52.7)	499		1	
Yes	110 (38.7)	174 (61.3)	284	0.011	1.63	1.12–2.38
COVID-19 related experiences						
No known diagnosis of COVID-19	255 (42.6)	344 (57.4)	599		1	
Tested positive for COVID-19	24 (38.1)	39 (61.9)	63	0.582	1.22	0.60–2.48
Tested negative for COVID-19 but self-isolating	48 (51.6)	45 (48.4)	93	0.067	0.59	0.34–1.04
Recent overseas travel history and was in quarantine	8 (47.1)	9 (52.9)	17	0.593	0.72	0.22–2.38

## Data Availability

The datasets used and/or analyzed during the current study are available from the corresponding author upon reasonable request.

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
