# Peer review of "COVID-19: Factors Associated with the Psychological Distress, Fear and Resilient Coping Strategies among Community Members in Saudi Arabia"

_healthcare, 2023, doi:10.3390/healthcare11081184_

Round 1

Reviewer 1 Report

Dear authors

This study examined factors associated with psychosocial distress, fear of COVID-19 and coping strategies amongst the general population in Saudi Arabia, and defined high-risk groups impacted by the pandemic.

It is important to identify the risky groups affected by the covid-19 pandemic, which negatively affects the whole world in many respects, but the study has important limitations. In my opinion, the most important methodological problem of the study is to analyze the three independent variables separately, but to report the results together. The fact that psychosocial distress, fear of COVID-19 and coping strategies are variables that are related to each other, that there is no analysis related to this in the study and that it is not mentioned is an important shortcoming. Parallel to this situation, the discussion part is insufficient.

Abstract: It is appropriate to express the results of the study more clearly. furthermore, recommendations should be limited in the light of these results.

Introduction: Adequate and appropriately written.

Method , result and discussion: Psychosocial distress, fear of COVID-19 and coping strategies should be analyzed separately, their relationship with each other should be evaluated, and I think it would be appropriate to determine risk factors and risky groups over an outcome variable. I think it would be appropriate to restructure the discussion section after this evaluation.

Tables: I am of the opinion that the number of tables should be reduced.

Author Response

We would like to sincerely thank you for the opportunity to revise our manuscript. We appreciate the feedback and constructive suggestions provided by the reviewer. It is our belief that the manuscript has improved due to the modifications made.

The manuscript has been revised to address the comments from the reviewer. Changes to the manuscript are marked using track changes and a response document has outlined how each comment was addressed.

Reviewer 2 Report

The article is interesting. Once again, the topic of Covid-19 was taken up. My note: the results are summarized as a reflection of the tool. The results dominate the analysis. The authors should analyze the research results. The next step is discussion. The authors attempt to compare the research results of other scientists. They do it very poorly. Many such studies have been conducted. They can make comparisons. This text needs to be corrected.

Author Response

(The authors gave the same response as above.)

Reviewer 3 Report

Dear authors, 

the paper is interesting and original. The main purpose is to explore the factors associated with the psychological distress, fear and resilient coping strategies among Community Members in Saudi Arabia during the COVID-19. In my opinion, however, there are some parts that should be expanded to better explain the topinc analyzed. Firts, in the abstract I would add the tools used to collect the data. Second, the introduction is too short: it is necessary to report the studies previously conducted in the literature with respect to the issues analyzed in order to support the theoretical framework of the study carried out. In the methodology section, it is necessary to describe in detail the tools used (number of items, sub-dimensions, etc.). Finally, I suggest you create a separate section for the description of the limits of your search. I hope my suggestions can improve the quality of your article. 

Author Response

(The authors gave the same response as above.)

Reviewer 4 Report

COVID-19 caused the worst international public health crisis, accompanied by major global economic downturns, mass-scale job losses, which impacted on the psychosocial wellbeing of the worldwide population including Saudi Arabia. Evidence on the high-risk groups impacted by the pandemic was non-existent in Saudi Arabia.

Authors examined factors associated with psychosocial distress, fear of COVID-19 and coping strategies amongst the general population in Saudi Arabia.

They proposed a cross-sectional study was conducted using an anonymous online questionnaire during Dec-2020 to Jan-2021.

The study highlighted that people in Saudi Arabia were at a higher risk of psychosocial distress and fear along with low resilience during the COVID-19 pandemic, warranting urgent attention from healthcare providers and policymakers, to provide specific mental health support strategies for their wellbeing currently and to avoid a post-pandemic mental health crisis.

The manuscript is interesting.

It needs some improvements before being considered for acceptance.

Strengths

-The study is written with enthusiasm and passion.

-Many interesting and worthy of publication data are reported

Points of weakness

- It is necessary to pay a little more attention to the presentation aspects of the results to avoid that the reader loses the message

- Greater attention to the editorial aspects would improve the quality of the manuscript

Further comments:

1. The abstract needs to be revised. It must be blunt. Numerical data must be minimized.

2. The purpose must be better explained and inserted before the methods.

3. Avoid one-sentence paragraphs. See the par. 2.1.

4. I suggest to insert the first part of results in par. 2.3 (it is the categorization of the participants).

5. Introduces the themes of the results arranged into paragraphs.

6. Did the submission of the questionnaire use electronic tools (e.g. google froms, ms forms)? If yes explain ad details the process.

Author Response

We would like to sincerely thank you for the opportunity to revise our manuscript. We appreciate the feedback and constructive suggestions provided by the reviewers. It is our belief that the manuscript has improved due to the modifications made.

The manuscript has been revised to address the comments from each reviewer. Changes to the manuscript are marked using track changes and a response document has outlined how each comment was addressed.

Round 2

Reviewer 1 Report

Dear authors

Thank you for the corrections you made taking into account my comments. but I request you to reconsider my comment, which I think is methodologically important. I think that the COVID-19: Factors Associated with the Psychological Distress, Fear and Resilient Coping Strategies parameters, which I want to emphasize, should not be correlated with each other, but rather their effects should be examined with a further analysis such as regression analysis.

Kind regards

Author Response

Dear Reviewer

We value the recommendation made by the expert reviewer. We discussed amongst authors and decided to omit the analyses examining the correlation between tools. Therefore, our paper just focused on identifying factors associated with each of those three outcomes: psychological distress, fear and coping. To identify those factors, we have used both univariate and multivariate logistic regression analyses.

Once again we would like to sincerely thank you for the opportunity to revise our manuscript

Reviewer 3 Report

Dear Authors, 

I am pleased that you have edited the paper as directed. I think this new version of the article is clearer and more complete. 

I suggest reviewing the format and the text editing (spaces, writing format). 

Author Response

Dear  reviewer

We would like to thank the reviewer for the kind appreciation. We have reviewed and updated the manuscript accordingly for format and the text editing. 

Once again we would like to sincerely thank you for the opportunity to revise our manuscript